

# Estimating the one-repetition maximum on the leg-press exercise in female breast cancer survivors

David M. Díez-Fernández[1,2], Andrés Baena-Raya[1,2], Amador García-Ramos[3,4], Alba Esteban-Simón[1,2], Manuel A. Rodríguez-Pérez[1,2], Antonio J. Casimiro-Andújar[1,2] and Alberto Soriano-Maldonado[1,2]

[1] Department of Education, Faculty of Education Sciences, University of Almería, Almería, Spain
[2] SPORT Research Group (CTS-1024), CERNEP Research Center, University of Almería, Almería, Spain
[3] Department of Physical Education and Sport, Faculty of Sport Science, University of Granada, Granada, Spain
[4] Department of Sports Sciences and Physical Conditioning, Faculty of Education, Universidad Católica de la Santísima Concepción, Concepción, Chile

Corresponding author
Amador García-Ramos,
amagr@ugr.es

## ABSTRACT

We examined the accuracy of twelve different velocity-based methods for predicting the bilateral leg-press exercise one-repetition maximum (1RM) in breast cancer survivors. Twenty-one female breast cancer survivors (age 50.2 ± 10.8 years) performed an incremental loading test up to the 1RM. Individual load-velocity relationships were modeled by linear and quadratic polynomial regression models considering the mean velocity (MV) and peak velocity (PV) values recorded at five incremental loads (~45-55-65-75-85% of 1RM) (multiple-point methods) and by a linear regression model considering only the two distant loads (~45–85% of 1RM) (two-point method). The 1RM was always estimated through these load-velocity relationships as the load associated with a general (MV: 0.24 m/s; PV: 0.60 m/s) and an individual (MV and PV of the 1RM trial) minimal velocity threshold (MVT). Compared to the actual 1RM, the 1RMs estimated by all linear regression models showed trivial differences (Hedge's g ranged from 0.08 to 0.17), very large to nearly perfect correlations (r ranged from 0.87 to 0.95), and no heteroscedasticity of the errors (coefficient of determination ($r^2$) < 0.10 obtained from the relationship of the raw differences between the actual and predicted 1RMs with their average value). Given the acceptable and comparable accuracy for all 1RM linear prediction methods, the two-point method and a general MVT could be recommended to simplify the testing procedure of the bilateral leg-press 1RM in breast cancer survivors.

## INTRODUCTION

Breast cancer is the most common type of cancer in women (*Sancho-Garnier & Colonna, 2019*). Thanks to the improvements in early diagnosis and treatment, specifically as of

2018, the death rate has decreased from its peak (1989) for female breast cancer by 41% (*Siegel et al., 2021*). However, an increasing number of women deal with the consequences of the disease and its treatments (*Campbell et al., 2012*). Among the side effects of breast cancer treatments, sarcopenia (the loss of muscle mass) and the loss of muscular strength (approximately 25% lower in maximal voluntary isometric contraction and maximal isokinetic peak torque in knee flexors and extensors compared to healthy individuals; *Klassen et al., 2017*) are commonly observed. Considering that muscular strength is an important prognostic factor in cancer survivors (*García-Hermoso et al., 2018*) that can be enhanced through resistance training (RT) (*Soriano-Maldonado et al., 2022*), it is important to optimize the testing procedures for the assessment of muscular strength in this population.

Quantifying and monitoring the relative load during RT is essential, since the load lifted is one of the variables that most influences training adaptations (*Fry, 2004*). Maximal dynamic strength is frequently assessed *via* a one-repetition maximum (1RM) protocol, as seen in recent studies involving breast cancer survivors (*Murri et al., 2023*; *Garcia-Unciti et al., 2023*). However, the 1RM value can increase rapidly due to RT in untrained individuals (*Garcia-Ramos & Jaric, 2018*; *González-Badillo & Sánchez-Medina, 2010*) such as, potentially, breast cancer patients. Therefore, the 1RM should be frequently assessed if practitioners want to accurately prescribe the loads relative to the individual's maximal dynamic strength capacity. Nevertheless, traditional 1RM testing protocols require the individual to perform either a maximal lift or repetitions to failure against a submaximal load which induce excessive mechanical and metabolic strain and substantial fatigue (*Izquierdo et al., 2006*). Moreover, traditional protocols also increase blood pressure and stress on the muscles, bones, and connective tissues which could increase the risk of musculoskeletal injuries (*Brzycki, 1993*). This could be especially relevant in breast cancer patients, as fatigue is a core cancer-related symptom (*Stasi et al., 2003*). Therefore, alternative approaches to precisely estimate the 1RM minimizing the testing-induced fatigue are needed in this population.

Movement velocity of the concentric phase can be used to estimate the 1RM in several exercises, so that patients do not need to perform repetitions to failure or lift a maximal load (*Banyard, Nosaka & Haff, 2017*; *Benavides-Ubric et al., 2020*; *García-Ramos et al., 2019*). Although the 45° inclined bilateral leg-press exercise is one of the most commonly used exercises in fitness centers, only three studies have reported the load-velocity relationship in this exercise in trained male athletes (*Conceição et al., 2015*), in older women (*Marcos-Pardo et al., 2019*), and in female breast cancer survivors (*Díez-Fernández et al., 2021*). However, to our knowledge, no study has investigated the accuracy of individualized load-velocity relationships to predict the 1RM in the 45° inclined bilateral leg-press exercise. Given the feasibility and safety that the bilateral leg-press exercise offers to breast cancer patients to improve strength and functionality during the follow up of adjuvant therapy (*Cešeiko et al., 2020*), exploring 1RM prediction in this exercise could improve the efficiency of maximal dynamic strength assessment and RT prescription in this population.

Mean velocity (MV) and peak velocity (PV) variables have been used to determine the load-velocity relationship and predict the 1RM in different exercises (*Garcia-Ramos & Jaric, 2018*; *Benavides-Ubric et al., 2020*; *Díez-Fernández et al., 2021*). Although the general load-velocity relationship has been traditionally determined through polynomial regression models (*González-Badillo & Sánchez-Medina, 2010*; *Sánchez-Medina et al., 2014*), recent evidence suggests that linear models could be more appropriate to estimate the 1RM through the individualized load-velocity relationship (*Banyard, Nosaka & Haff, 2017*; *Ruf, Chéry & Taylor, 2018*). Briefly, linear models assess movement velocity against two (two-point method) or more than two (multiple-point method) submaximal loads (*Garcia-Ramos & Jaric, 2018*; *García-Ramos et al., 2019*). Regardless of the number of loads tested, the 1RM is always predicted as the load associated with the velocity of the 1RM trial (minimal velocity threshold; MVT). Some studies have used an individualized MVT for each individual (*Banyard, Nosaka & Haff, 2017*; *Jukic et al., 2020*), but recent literature suggests that a general MVT can predict the 1RM with comparable precision (*Benavides-Ubric et al., 2020*; *García-Ramos et al., 2019*; *Janicijevic et al., 2021*). However, in the specific case of the bilateral leg press exercise, it is unknown whether the velocity variable (MV *vs* PV), regression model applied (linear *vs* quadratic polynomial), number of loads tested (two or more), or velocity value used as the MVT (individual 1RM velocity *vs* average across the subjects 1RM velocity) can affect the accuracy of the 1RM prediction. This is important because some of these factors have shown to influence the accuracy in the prediction of the 1RM in other exercises such as squat (*Caven et al., 2020*) and bench press (*Janicijevic et al., 2021*).

Therefore, the aim of this study was to compare the accuracy of twelve 1RM prediction methods based on the combination of two velocity variables (MV *vs* PV), three regression models (multiple-point linear *vs* two-point *vs* multiple-point polynomial) and 2 MVT (individual *vs* general) during the leg-press exercise in female breast cancer survivors. We hypothesized that a) MV and PV would be equally valid to estimate the 1RM in the leg-press exercise (*Conceição et al., 2015*), b) the linear regression models would provide good accuracy in the 1RM prediction (*Janicijevic et al., 2021*), and c) the error in the 1RM prediction would not significantly differ between the 2 MVTs (*Jukic et al., 2020*; *Janicijevic et al., 2021*; *Caven et al., 2020*).

# MATERIALS AND METHODS

## Participants

As part of the EFICAN study (*Soriano-Maldonado et al., 2019*), a group of 24 women volunteered to take part in this study. A sample size of 20 women has been determined based on previous regression analyses (*Díez-Fernández et al., 2021*). The required sample sizes was calculated using z-score for the 95% confidence interval and with a minimum power of 80%. All participants had experienced surgical intervention and had successfully completed primary treatment for breast cancer within the preceding 10 years. The participants' characteristics are presented in Table 1. The exclusion criteria included being scheduled for breast reconstruction in the following 3 months, having any comorbidity that might contraindicate the performance of a maximum test or not reaching a MV above

**Table 1 Descriptive characteristics of the study sample (mean ± SD).**

| | | |
|---|---|---|
| Subjects' physical characteristics | Age (years) | 50.2 ± 10.8 |
| | Mass (kg) | 69.6 ± 15.2 |
| | Height (cm) | 160.5 ± 5.3 |
| | Body mass index (kg/m$^2$) | 27.5 ± 6.8 |
| | Body fat mass (kg) | 26.4 ± 12.36 |
| | Body mucle mass (kg) | 23.7 ± 3.4 |
| | 1RM bilateral leg-press (kg) | 117.4 ± 24.8 |
| | 1RM bilateral leg-press (normalized per kg of body mass) | 1.69 ± 0.32 |
| Treatment | Chemotherapy (sessions) | 7.7 ± 3.9 |
| | Radiotherapy (sessions) | 26.4 ± 6.1 |
| Medical information | Tumor type, HR+HER2-/HR+HER2+/HR-HER2+/HR-HER2-, (%) | 65.1/18.3/3.3/13.3 |
| | Surgical procedure, n (%) Tumorectomy/Mastectomy | 15 (71.4)/6 (28.6) |
| | Lymph node resection, n (%) | 9 (42.9) |
| | Lymphedema, n (%) | 2 (9.5) |
| | Endocrine therapy, n (%) | 18 (85.7) |

**Note:**
Data are mean ± standard deviation. 1RM, one-repetition maximum; HR, hormone receptor; HER2, human epidermal growth factor receptor 2.

0.80 ms$^{-1}$ with the minimum load (*i.e.*, 25 kg) during the familiarization session. The present research was approved by the Ethics Committee of the University of Almería, Spain (ref: UALBIO2022/008). After being informed of the purpose of the study and the experimental procedures, the participants signed a written informed consent form prior to participation.

## Study design

A descriptive cross-sectional study was conducted to assess the most accurate method for predicting the 1RM during the bilateral leg-press exercise in female breast cancer survivors. For this, each participant underwent a single test performing an incremental loading test up to the 1RM and the movement velocity of all loads was recorded to determine the individual load-velocity relationship. Subjects underwent a preliminary session during which they were familiarized with the measuring instruments and exercise protocol. During this session, the subjects performed an incremental loading test (2–4 loads) from 25 kg until reaching a MV of approximately 0.70 m/s. The researchers checked the technique and encouraged them to move the loads at the maximum intended velocity during the concentric phase. The analyses were performed considering twelve different velocity-based methods (2 velocity variables × 3 regression models × 2 MVTs) to predict the 1RM. The velocity variables were MV and PV. The incremental loading test consisted of 10.3 ± 2.1 loads. We used five loads (~45%, 55%, 65%, 75% and 85% 1RM) to estimate the 1RM for the multiple-point method (linear and quadratic polynomial regression equations), whereas only the two distant loads (~45% and 85% 1RM) were used for the two-point method. The 1RM was determined for each model as the load associated with a

general (0.24 m/s for MV and 0.60 m/s for PV) or individual (velocity of the 1RM trial) MVT.

## Testing procedures

Participants attended a previous medical examination to evaluate the presence of any restrictions for performing a maximum strength test. Moreover, body height (digital-Seca 202 stadiometer; Seca Ltd, Hamburg, Germany), weight and body composition (electrical bioimpedance-InBody 120; InBody Co Ltd, Seoul, South Korea) were assessed.

A standardized warm-up was performed at the beginning of the test including 5 min of walking, 2 min of lower-limb dynamic mobility, 30 s performing body weight squats and a set of six repetitions with a 25 kg load in the bilateral leg-press exercise. The initial load during the test was set at 25 kg for all the subjects. Similar to a previous study (*Marcos-Pardo et al., 2019*) the load was gradually increased, initially in 20 kg increments until reaching a MV of ~0.90 m/s. Subsequently, there were 10 kg increments until reaching a MV of ~0.50 m/s. Starting at this MV, the load was increased by 5-, 2.5- or 1 kg in consensus with the subjects until reaching the 1RM. The last load that was correctly lifted was determined as the 1RM value. Like a previous study (*Conceição et al., 2015*), during the incremental loading test, subjects performed three repetitions with low loads (>0.90 m/s), 2 with medium loads (0.90–0.60 m/s) and only 1 with high loads (<0.60 m/s). The recovery time between sets was of 3 min for low loads, 4 min for medium loads, and 5 min for high loads. Movement velocity during the concentric phase of all repetitions was recorded with a linear velocity transducer (T-Force System; Ergo-Tech, Murcia, Spain). Strong verbal encouragement was provided during testing to motivate subjects to give maximal effort. Only the repetition with the highest maximum velocity for each load was considered for further analysis.

During the test, to be considered as a valid repetition, the knee flexion had to reach a ~90° during the eccentric phase and finish the movement reaching a complete knee extension. This position was recorded and marked for each subject, and an audible signal was provided by the evaluator when the participant reaching the ~90° of knee flexion. A momentary pause (~1 s) was imposed between the eccentric and concentric phases. If the execution did not meet the technique criteria or the displacement range was deemed inadequate (at the evaluator's discretion), a new set with the same absolute load was performed after the corresponding rest period. Both the leg press features and the complete execution of the exercise are clearly described elsewhere (*Díez-Fernández et al., 2021*).

## Statistical analysis

Descriptive data are presented as means ± standard deviations. A three-way repeated-measures analysis of variance (ANOVA) (velocity variable (MV *vs* PV) × regression model (multiple-point linear *vs* two-point *vs* multiple-point polynomial) × MVT (general *vs* individual)) was applied on the absolute differences between the actual and predicted bilateral leg-press 1RMs. The Greenhouse-Geisser correction was applied when the sphericity assumption was violated ($p < 0.05$). In addition, the validity of the 1RM prediction methods with respect to the actual 1RM was examined through effect size (ES,

95% confidence interval) using Hedge's g (*Hedges & Olkin, 1985*), the raw differences (kg), and the Pearson's correlation coefficients (*r*, 95% confidence interval).

The heteroscedasticity of the errors ($r^2$; coefficient of determination obtained from the relationship of the raw differences between the actual and predicted 1RMs with their average value) was also quantified to determine the existence of proportional bias. The ES magnitude was interpreted using the subsequent scale: *trivial* (<0.20), *small* (0.20–0.59), *moderate* (0.60–1.19), *large* (1.20–2.00) and *very large* (>2.00) (*Hopkins et al., 2009*). Qualitative interpretations of the *r* coefficients were defined as follows: *trivial* (0.00–0.09), *small* (0.10–0.29), *moderate* (0.30–0.49), *large* (0.50–0.69), *very large* (0.70–0.89), *nearly perfect* (0.90–0.99), *perfect* (1.00) (*Hopkins et al., 2009*). Heteroscedasticity of error was defined as a $r^2 > 0.10$ (*Atkinson & Nevill, 1998*). Alpha was set at 0.05. All statistical analyses were performed using SPSS software (version 26; IBM SPSS, INC., Chicago, IL, USA).

## RESULTS

Of the 25 participants who offered to take part, 21 effectively completed the test and are documented in this manuscript. Among them, two participants expressed discomfort in their lower back, while one reported pressure on the breast prosthesis during the testing process. Consequently, they preferred not to continue their participation and were excluded from the study. One participant was excluded from the study because she failed to execute at maximum intended velocity during all the loads of the incremental test.

The absolute errors for the different velocity variables, regression models, and MVTs are depicted in Fig. 1. The ANOVA revealed a significant main effect of the regression model (F = 4.96, $p$ = 0.034) with the multiple-point polynomial (12.0 ± 12.7 kg) showing greater absolute errors than the multiple-point linear (6.9 ± 5.4 kg) and two-point (6.8 ± 4.5 kg). The main effects of the velocity variable (F = 0.12, $p$ = 0.729) and MVT (F = 0.66, $p$ = 0.428) did not reach statistical significance. Similarly, the interactions velocity variable × regression model (F = 2.36, $p$ = 0.138), velocity variable × MVT (F = 2.20, $p$ = 0.153), and regression model × MVT (F = 1.95, $p$ = 0.172) neither reached statistical significance. Finally, the interaction velocity variable × regression model × MVT (F = 4.13, $p$ = 0.041) reached statistical significance. The triple interaction was caused because the multiple-point polynomial and individual MVT were more accurate using PV than MV. Conversely, both linear regression models (multiple-point and two-point) and the general MVT were more accurate using MV than PV.

For all 1RM linear prediction methods, a trivial effect size (ES ranged from 0.08 to 0.17) and nearly perfect correlations (r ranged from 0.87 to 0.95) were observed. Was not observed heteroscedasticity of the errors (*i.e.*, $r^2 > 0.10$) within the multiple-point linear or two-point models. However, multiple-point polynomial using MV showed a small effect size (ES 0.22 and 0.28 using individual V1RM and general V1RM, respectively) and heteroscedasticity of the errors. In general terms an overestimation of the 1RM was observed for the 1RM prediction methods using MV (ranged from 1.95 to 9.49 kg) and underestimation using PV (ranged from 0.03 to 3.70 kg), except for multiple-point polynomial using the individual MVT that showed an overestimation of 1.07 kg (Table 2).

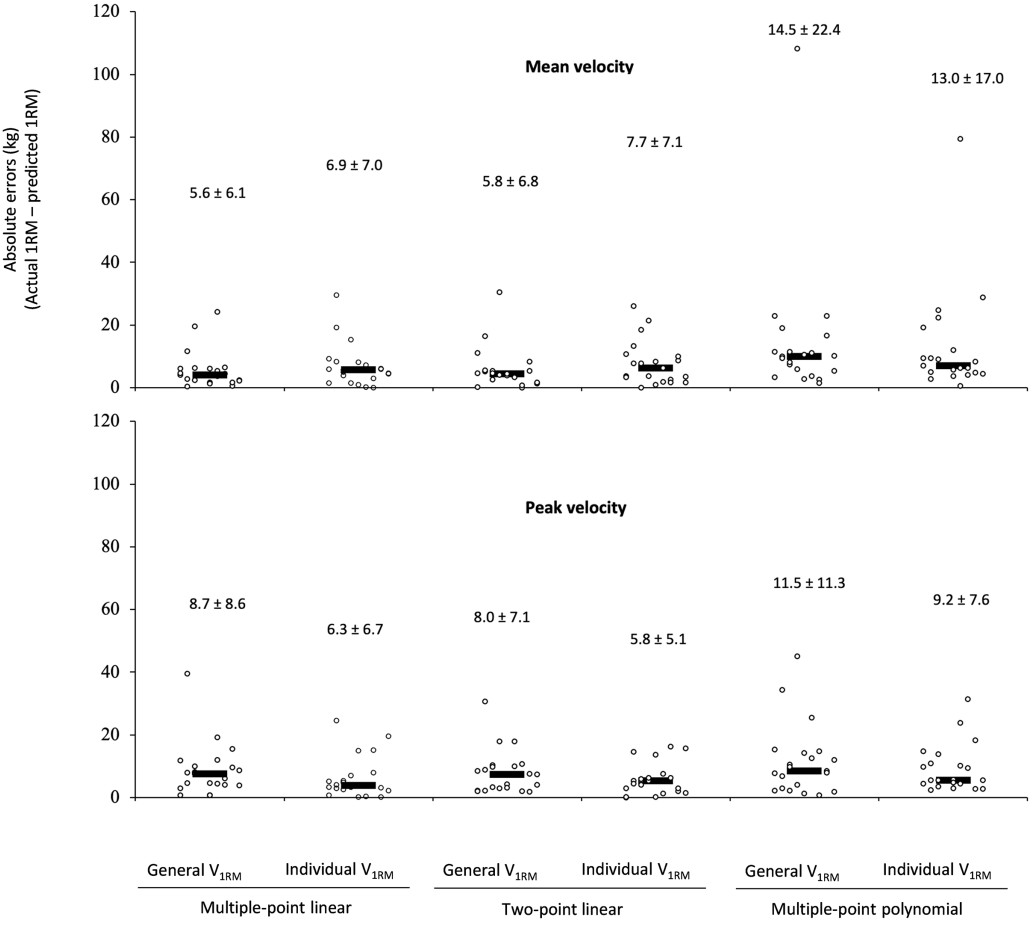

**Figure 1 Absolute differences between the actual one-repetition maximum (1RM) and the 1RM estimated using the different prediction methods during the leg-press exercise.** The black rectangle denotes the median value, while the circles represent individual data points. Numbers indicate the mean ± standard deviation.

## DISCUSSION

The main findings revealed that (a) MV, and PV revealed comparable precision; (b) both 1RM linear prediction methods were capable of estimating the 1RM with an acceptable and comparable precision regardless of the velocity variable, while the multiple-point polynomial was more accurate using PV than MV; (c) the individual MVT and general MVT revealed comparable precision. Therefore, regardless of the velocity variable and MVT, 1RM linear prediction methods are a valid alternative for obtaining a quick and precise estimation of the 1RM during the bilateral leg-press exercise in female breast cancer survivors.

Although estimating the 1RM without excessive effort is especially relevant in breast cancer patients since fatigue is a core cancer-related symptom (*Stasi et al., 2003*), to the best of our knowledge, this is the first study using velocity-based methods for the 1RM estimation in this population. As firstly hypothesized, the MV and PV were valid to estimate the 1RM during the leg-press exercise in female breast cancer survivors, which

**Table 2 Comparison of the directly measured bilateral leg-press one-repetition maximum (117.4 ± 24.8) with the 1RM estimated using different velocity variables, regression models, and minimal velocity thersholds.**

| Velocity | Regression models | MVT | Predicted 1RM (kg) | Raw Diff (kg) | ES [95% IC] | r [95% IC] | r² |
|---|---|---|---|---|---|---|---|
| MV | Multiple-point linear | General $V_{1RM}$ | 119.3 ± 24.6 | 1.96 ± 8.10 | 0.08 [−0.53 to 0.68] | 0.95 [0.87–0.98] | 0.004 |
| | | Individual $V_{1RM}$ | 119.3 ± 23.2 | 1.95 ± 9.76 | 0.08 [−0.53 to 0.68] | 0.92 [0.80–0.97] | 0.008 |
| | Two-point | General $V_{1RM}$ | 121.5 ± 25.7 | 4.09 ± 7.99 | 0.16 [−0.45 to 0.77] | 0.95 [0.89–0.98] | 0.044 |
| | | Individual $V_{1RM}$ | 121.6 ± 24.4 | 4.19 ± 9.70 | 0.17 [−0.44 to 0.77] | 0.92 [0.81–0.97] | 0.001 |
| | Multiple-point polynomial | General $V_{1RM}$ | 126.9 ± 40.3 | 9.49 ± 25.01 | 0.28 [−0.33 to 0.89] | 0.82 [0.59–0,92] | 0.462 |
| | | Individual $V_{1RM}$ | 124.4 ± 35.6 | 6.99 ± 20.42 | 0.22 [−0.38 to 0.83] | 0.83 [0.63–0.93] | 0.344 |
| PV | Multiple-point linear | General $V_{1RM}$ | 113.7 ± 20.8 | −3.70 ± 11.76 | −0.16 [−0.76 to 0.45] | 0.87 [0.71–0.95] | 0.084 |
| | | Individual $V_{1RM}$ | 114.2 ± 22.1 | −3.15 ± 8.68 | −0.13 [−0.74 to 0.47] | 0.93 [0.84–0.97] | 0.057 |
| | Two-point | General $V_{1RM}$ | 115.3 ± 20.9 | −2.08 ± 10.61 | −0.09 [−0.70 to 0.52] | 0.90 [0.76–0.96] | 0.097 |
| | | Individual $V_{1RM}$ | 115.9 ± 22.6 | −1.44 ± 7.69 | −0.06 [−0.67 to 0.54] | 0.95 [0.87–0.98] | 0.041 |
| | Multiple-point polynomial | General $V_{1RM}$ | 117.4 ± 24.5 | −0.03 ± 16.29 | 0.00 [−0.60 to 0.60] | 0.78 [0.52–0.90] | 0.001 |
| | | Individual $V_{1RM}$ | 118.5 ± 27.1 | 1.07 ± 12.03 | 0.04 [−0.56 to 0.65] | 0.90 [0.76–0.96] | 0.065 |

**Note:**
Data are mean ± standard deviation. MV, mean velocity; PV, peak velocity; RM, one-repetition maximum; MVT, minimal velocity thresholds; Raw diff, raw differences (Predicted 1RM–Actual 1RM); ES, Hedge's g effect size with 95% confidence intervals ((Predicted 1RM–Actual 1RM)/SD both); r, Pearson's correlation coefficient with 95% confidence intervals between actual RM and predicted 1RM; r², coefficient of determination obtained from the relationship of the raw differences between the actual and predicted 1RMs with their average value; $V_{1RM}$, velocity of the one-repetition maximum.

concurs with previous studies that analyzed in other populations the leg-press, half-squat, full-squat (*Conceição et al., 2015*) or the bench press (*García-Ramos et al., 2018*). Specifically, the methods using MV overestimated the 1RM from 1.95 to 9.49 kg, whereas the methods using PV showed an underestimation ranging from 0.03 to 3.70 kg (except multiple-point polynomial using the individual MVT that showed an overestimation of 1.07 kg). Interestingly, previous studies also showed an overestimation of the 1RM during the back squat when using the MV (*Banyard, Nosaka & Haff, 2017*). Similarly, it was reported that the predicted 1RM using MV is generally overestimated by 5–10 kg in the deadlift when compared to the actual 1RM (*Ruf, Chéry & Taylor, 2018*; *Jukic et al., 2020*). Therefore, it could be reasonable to use a higher MVT to minimize the overestimation of the 1RM when the individualized load-velocity relationship is modelled considering MV values in these exercises.

As hypothesized, the multiple-point linear and two-point were the most precise models for 1RM prediction during the leg-press exercise in female breast cancer survivors regardless of the velocity variable or MVT type. In this line, previous studies confirmed that the linear prediction methods should be preferably used for 1RM estimation in a variety of exercises (*García-Ramos et al., 2019*; *Janicijevic et al., 2021*; *Muñoz-López et al., 2017*). Regarding the number of loads, our current results concur with previous findings that demonstrate a comparable precision in the estimation of the 1RM between the linear multiple-point and two-point methods (*Garcia-Ramos & Jaric, 2018*; *Jukic et al., 2020*; *Janicijevic et al., 2021*). Note that both linear prediction methods showed nearly perfect correlations (r ranged from 0.87 to 0.95) between the actual and predicted 1RMs, and the absolute errors were very similar between multiple-point linear (6.9 ± 5.4 kg) and

two-point methods (6.8 ± 4.5 kg). Therefore, conducting the two-point method may be of practical interest to estimate the 1RM in the bilateral leg-press exercise when working with female breast cancer survivors, resulting in a quicker and less fatiguing testing procedure, as reported in a previous study using this method (*Garcia-Ramos & Jaric, 2018*).

Our third hypothesis was that the errors of the regression models for predicting the 1RM would not differ between the 2 MVTs. In this regard, *Jukic et al. (2020)* and *Janicijevic et al. (2021)* also failed to show significant differences between the individual and general V1RM for the accuracy in the prediction of the 1RM during the bench press and the deadlift exercises, respectively. For the squat exercise, a previous study (*Caven et al., 2020*) reported that the general MVT provided greater absolute errors (from 7.8 to 9.7 kg) than the individual MVT (from 4.9 to 6.3 kg), but these differences did not reach statistical significance. These results suggest that using a general MVT is valid and would avoid the need to perform a direct assessment of the 1RM, which is especially relevant when working with breast cancer survivors.

Several limitations of this study must be acknowledgment. First, there are two factors that could compromise the generalizability of the current findings to all female breast cancer survivors: (1) the inclusion of women who had undergone breast cancer surgery and completed the core treatments up to 10 years prior to enrollment, leading to a heterogeneous sample; (2) the lack of information on the history of hormone therapy, which holds significance due its potential impact on muscle strength. Second, despite multiple loads (10.3 ± 2.1) were executed until reaching the actual 1RM, only two distant loads (~45% and 85% 1RM) were used to estimate the 1RM through the two-point method. Thus, participants would likely be prepared to exert a maximum effort rather than executing only two loads. Therefore, it is important that when implementing the two-point method in practice, participants firstly preform a proper warm-up before recording the velocity performance against the two selected loads. Finally, our current study did not examine whether the two-point method reduces fatigue perception compared to multiple trials, as expected. To gain deeper insights into its practical implications, future research should address this question.

## CONCLUSIONS

As most important findings, we provide a new approach based on movement velocity to safely monitor the intensity during resistance training in breast cancer survivors. Thus, coaches or researchers can estimate the 1RM during bilateral leg-press exercise without using heavy loads that accumulate excessive fatigue in this population. The two-point and multiple-point linear prediction methods estimated the 1RM with acceptable and comparable precision regardless of the velocity variable or MVT type, while the multiple-point polynomial was more accurate using individual MVT and PV. From a practical perspective, authors recommend coaches and researchers to firstly determine the individual load-velocity relationship and then, using the two-point method (~45–85% of 1RM), MV and the load associated with a general MVT of 0.24 m/s to daily quantify the intensity during resistance training sessions. Importantly, this methodology allows one strength and conditioning coach or researcher to evaluate few patients at the same time.

## ACKNOWLEDGEMENTS

The authors thank our colleagues who collaborated with data collection, Celia Alcaraz-Garcia and Carlos Martinez-Rubio.

### Funding

This work was funded by the Patronato Municipal de Deportes, Ayuntamiento de Almería, and by the UAL Transfiere Research Program of the University of Almería [reference number: TRFE-SI-2019/004 and TRFE-SI-2022/010]. David M. Díez-Fernández was funded by a scholarship from the UAL Transfiere Research Program of the University of Almería [reference number: TRFE-BT-2019/002]. David M. Díez-Fernández and Andrés Baena-Raya are currently funded by the Ministry of Science, Innovation and Universities of the government of Spain (grant number: FPU19/04608 and FPU20/05746, respectively). Alba Esteban-Simon is currently funded by the Sede provincial de Almería de la Asociación Española Contra el Cáncer and the AECC Scientific Foundation [PRDAM222381ESTE]. The funders had no role in study design, data collection and analysis, decision to publish, or preparation of the manuscript.

### Grant Disclosures

The following grant information was disclosed by the authors:
Patronato Municipal de Deportes, Ayuntamiento de Almería, University of Almería: TRFE-SI-2019/004, TRFE-SI-2022/010.
University of Almería: TRFE-BT-2019/002.
Ministry of Science, Innovation and Universities of the Government of Spain: FPU19/04608 and FPU20/05746.
Sede Provincial de Almería de la Asociación Española Contra el Cáncer and the AECC Scientific Foundation: PRDAM222381ESTE.

### Competing Interests

Amador García-Ramos is an Academic Editor for PeerJ. The authors declare they have no conflict of interest or professional relationships with companies or manufacturers that might benefit from the results of this study.

### Author Contributions

- David M. Díez-Fernández conceived and designed the experiments, performed the experiments, analyzed the data, prepared figures and/or tables, authored or reviewed drafts of the article, and approved the final draft.
- Andrés Baena-Raya performed the experiments, authored or reviewed drafts of the article, and approved the final draft.
- Amador García-Ramos conceived and designed the experiments, analyzed the data, prepared figures and/or tables, authored or reviewed drafts of the article, and approved the final draft.

- Alba Esteban-Simón performed the experiments, authored or reviewed drafts of the article, and approved the final draft.
- Manuel A. Rodríguez-Pérez conceived and designed the experiments, performed the experiments, authored or reviewed drafts of the article, and approved the final draft.
- Antonio J. Casimiro-Andújar conceived and designed the experiments, authored or reviewed drafts of the article, project management, and approved the final draft.
- Alberto Soriano-Maldonado conceived and designed the experiments, authored or reviewed drafts of the article, project management, and approved the final draft.

## Human Ethics

The following information was supplied relating to ethical approvals (*i.e.*, approving body and any reference numbers):

The present research was approved by the Ethics Committee of the University of Almería, Spain (ref: UALBIO2022/008).

## Data Availability

The raw measurements are available in the Supplemental File.

## Supplemental Information

Supplemental information for this article can be found online at http://dx.doi.org/10.7717/peerj.16175#supplemental-information.

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
