# Peer review of "Estimating the one-repetition maximum on the leg-press exercise in female breast cancer survivors"

_PeerJ, doi:10.7717/peerj.16175_

## Round 0.1 · original submission · Minor Revisions

I commend the author for a well-structured article.

In addition to the reviewers’ comments, I have the following minor comments:
Line 53: Please, specify what ES you are using also in the abstract.
Line 54: Please, try to clarify how the heteroscedasticity was measured.
Line 81-83 and 322-324: Can you provide references regarding the superiority in terms of safety and fatiguing effect of the proposed methods compared to the classical 1-RM measurement method?
Line 93-94: I suggest the authors to rewrite this sentence in a more cautious and less affirmative way, while keeping the message of this sentence.
Line 130: Please, clarify which statistical test/indicator the 95% CIs are referring to.
Line 153-160: Please, try to make the study design clearer for the reader (e.g., specifying the number of testing days and the tests performed in each day).
Line 202: Please, clarify the reasons for choosing the absolute difference as the dependent variable. This will help the reader understand the importance of the RM-ANOVA’s results.
Line 206: Is there a reason why other indicators assessing agreement (e.g., CCC or Bland-Altman’s mean bias and LoA were not computed)? I think that including these indicators might further strengthen the results of the study.
Line 331-332: It is not clear to me the point that the authors are trying to make here, can you please clarify it?

Regards,
Carlo

·

Basic reporting

no comment

Experimental design

1. What is the rational of excluding those with more than 300 min of exercise per week? This could be any exercise unrelated to building strength and muscle.
2. Was depth monitored and controlled? Was there specific criteria in the technique controlled in any way? I was unable to access ref 15 to know.
3. Line 177: What do you mean by “reaching that individual position”?
4. Can you provide an image of the LPT. These are accurate only in the vertical direction. With a 45°, was it set at this angle?
5. With ~ 12 sets and multiple reps/sets, justify why you suggest fatigue was not a factor especially for this population. The NSCA recommends no more than 5 sets during 1RM testing.

Validity of the findings

1. Explain why you think that MV or PV was more accurate in one model compared to the other

Additional comments

The study was well written and designed.

Reviewer 2 ·

Basic reporting

INTRODUCTION:
General comment: The english language should be more professional and formal. The authors should implement the scientific evidence about their field of research that provide the background for their hypotesis. Moreover, they should avoid their personal opinion.

(Lines 68-70) Authors should avoid personal comments or conclusion in the background, but they should use references that explain the evidence/limitations in clinical practice for the assessment of muscle strength which will support their hypotesis.
(Lines 75-77) The authors should check whether there are recent scientific references which involve 1RM assessment in breast cancer patients. Moreover, Authors should use some studies as reference that use 1RM as prescription for physical exercise in breast cancer patients and obtenined positive results
(doi: 10.3390/cancers15010034)

(Lines 81-83)The authors should not use conclusions here, but should explain what alternative approaches can be used in breast cancer patients (ref.13, 14,15). Explain if it is possible, using scientific references, why this method might be less stressful for patients (if anyone has considerd these aspects).

Experimental design

DISCUSSION:
(Line 294): How the perception of fatigue was valuated? to give these conclusions, did the Authors assess also the perception of fatigue during the test?

Validity of the findings

no comment

Additional comments

(Line 72): Please explain the abbreviation if it Is the first time that you use.

---

## Round 0.2 · accepted · Accept

Dear authors,

I am pleased to inform you that the article is now suitable for publication in PeerJ.

Congrats,

Carlo

·

Basic reporting

No comment

Experimental design

no comment

Validity of the findings

no comment

Reviewer 2 ·

Basic reporting

the Authors have responded at comments of te first round revision, the article can be published on PeerJ

Experimental design

no comment

Validity of the findings

no comment

Additional comments

no comment